# Recovery of Platinum Group Metals from Leach Solutions of Spent Catalytic Converters Using Custom-Made Resins

Pulleng Moleko-Boyce * , Hlamulo Makelane *, Mbokazi Z. Ngayeka and Zenixole R. Tshentu *

Department of Chemistry, Nelson Mandela University, P.O. Box 77000, Port Elizabeth 6001, South Africa; s214077454@mandela.ac.za
* Correspondence: pulleng.moleko-boyce@mandela.ac.za (P.M.-B.); hlamulo.makelane@mandela.ac.za (H.M.); zenixole.tshentu@mandela.ac.za (Z.R.T.); Tel.: +27-41-504-2074 (Z.R.T.)

**Abstract:** Platinum group metals (PGMs) play a key role in modern society as they find application in clean technologies and other high-tech equipment. Spent catalytic converters as a secondary resource contain higher PGM concentrations and the recovery of these metals via leaching is continuously being improved but efforts are also directed at the purification of individual metal ions. The study presents the recovery of PGMs, namely, rhodium (Rh), platinum (Pt) and palladium (Pd) as well as base metals, namely, zinc (Zn), nickel (Ni), iron (Fe), manganese (Mn) and chromium (Cr) using leachates from spent diesel and petrol catalytic converters. The largest amount of Pt was leached from the diesel catalytic converter while the petrol gave the highest amount of Pd when leached with aqua regia. Merrifield beads (M) were functionalized with triethylenetetramine (TETA), ethane-1,2-dithiol (SS) and bis((1$H$-benzimidazol-2-yl)methyl)sulfide (NSN) to form M-TETA, M-SS and M-NSN, respectively, for recovery of PGMs and base metals from the leach solutions. The adsorption and loading capacities of the PGMs and base metals were investigated using column studies at 1 M HCl concentration. The loading capacity was observed in the increasing order of Pd to be 64.93 mmol/g (M-SS), 177.07 mmol/g (M-NSN), and 192.0 mmol/g (M-TETA), respectively, from a petrol catalytic converter. The M-NSN beads also had a much higher loading capacity for Fe (489.55 mmol/g) compared to other base metals. The finding showed that functionalized Merrifield resins were effective for the simultaneous recovery of PGMs and base metals from spent catalytic converters.

**Keywords:** spent catalytic converters; platinum group metals; base metals; recovery

## 1. Introduction

The production and the use of platinum group metals (PGMs) have increased globally in the past half-century resulting in high-tech applications [1]. Current ways to obtain raw materials include mining and metal processing [2,3] and recycling of spent products such as e-waste and catalytic converters [4,5]. The PGM industry has developed into being an exclusive supplier for the largest components of high-tech equipment and clean technologies over the years [6]. The development of high technology products uses PGMs due to their unique properties such as high electrical conductivity and catalytic activities as well as high corrosion and oxidation resistance [7]. Thus, the commercial use of precious metals has rapidly increased both in number and quantity resulting in improvements in the quality of life.

PGMs are becoming depleted and there are already limited resources of these precious metals as a result demand does not meet supply. It is necessary to recognize that the primary and secondary production of these metals is complementary and mutually dependent [8]. The secondary production of PGMs includes recycling which recovers metals from spent materials such as automotive catalysts [9], and electronic scraps [10] as well as residues created in primary production [11]. Secondary production contributes significantly to supply and demand. Therefore, the recycling of PGMs aims at incorporating metals into a

product or component back to the market at the end of the component's useful life [12] and PGMs recovery from end-of-life vehicles has increased over the years [13].

Recent studies have shown interest in the recovery of PGMs from spent catalytic converters [1,14–18] which consist of the catalytic material that contains a mixture of platinum (Pt), palladium (Pd), and rhodium (Rh) [17,19,20]. Most petrol and diesel vehicles, including automobiles, trucks, buses, trains, motorcycles, and planes, have exhaust systems employing a catalytic converter and Pt, Pd, and/or Rh are active components of PGMs that convert harmful gases emitted from vehicle engines to relatively harmless gases by both the reduction of nitrogen oxides ($NO_x$) into nitrogen $N_2$ and the oxidation of hydrocarbons and CO to $CO_2$ [21]. The cumulative PGMs concentration in an automotive catalyst ranges between 0.1% and 0.2%, [22,23] and in the commercial operations 95% of PGM recovery rates are achieved from a charge with very low concentrations of PGM (<0.1%) [21].

The recovery methods and their parameters have been reported, namely, pyrometallurgy [24], hydrometallurgy [25–27] and biometallurgy [26,27]. Hydrometallurgy uses oxidants such as aqua regia to dissolve PGMs, while also releasing considerable amounts of $NO_x$ gases [28,29]. The hydrometallurgical technology process has been widely used for PGMs recovery and dismantling is the most important step before the leaching step during the PGMs recovery [17,30,31]. Selective extraction and separation of PGMs from spent catalytic converters is a challenge in the subsequent hydrometallurgical processing [32] and developing extraction methods for the metal ions is required. Therefore, the development of coordination chemistry, based on the outer sphere or inner-sphere mechanism, for selective separation of Rh, Pt, Pd in chloride media has been of interest [33]. The extraction mechanism of the PGMs is essential for achieving the selectivity of extractants through the understanding of structural information of precious metals complexes in an aqueous chloride solution [34]. Chloride complexes for PGMs are well studied and allow excellent conditions for PGMs dissolution [27]. Therefore, chloride medium has been widely used in the hydrometallurgical recovery of PGMs due to higher leachability of complex metals, stability of chloride complexes and regeneration of leaching reagents [35].

The speciation in a chloride medium is critical for successful separations, 1 M HCl solutions platinum(IV) exists as the hexachlorido species ($[PtCl_6]^{2-}$) while palladium(II) is in the tetrachlorido form and rhodium(III) in lower chlorido forms, thus the aqua ligands on the rhodium(III) may be susceptible to ligand substitution despite the inert nature of rhodium(III). Possible rhodium(III) species formed in 1 M HCl matrix are the mer and fac-$[RhCl_3(H_2O)_3]$, trans-$[RhCl_4(H_2O)_2]^-$ and cis-$[RhCl_4(H_2O)_2]^-$. The substitutionally labile nature of $[PdCl_4]^{2-}$ also makes it susceptible to the inner sphere complexation mechanism while ($[PtCl_6]^{2-}$ is more susceptible to the outer sphere mechanism (ion-pairing). Therefore, the exploitation of the coordination chemistry of precious metal ions is critical in deriving successful separations and the reactivity of the PGMs depends on the oxidation state of the metal ion as well as the nature of the extracting ligands [36,37]. The order of reactivity with soft donor ligands is directly related to periodicity, with second-row precious metals being more reactive than the third-row metals. Metals in their divalent oxidation state are readily susceptible towards substitution by soft donor ligands and rates of substitution can be several orders of magnitude faster than for metals in their higher oxidation states [37]. Pt and Pd prefer soft donor ligands as opposed to σ donor only ligands such as aliphatic amines and ammonia but have a greater preference for π acceptor ligands such as sulphur, arsenic and phosphorous donors. The divalent palladium complexes are spin-paired square planar $d^8$ systems, while tetravalent rhodium ions are spin-paired octahedral $d^6$ systems [38,39]. Rhodium and iridium commonly occur as very stable trivalent states. These metals are primarily found in the form of spin-paired ($d^6$), kinetically inert octahedral complexes [40,41].

This work focuses mainly on the development of functional materials for the recovery of platinum group metals from leachates of spent catalytic converters. The application of ligands containing nitrogen (*N*) and/or sulphur (S) atoms have been explored. The following ligands were investigated, triethylenetetramine (TETA), ethane-1,2-dithiol (SS)

and *bis*((1*H*-benzimidazol-2-yl)methyl)sulfide (NSN) (Figure 1). Application of *N* and/or S ligands have been explored in complexation of base metals and PGMs [39–41]. The work was performed in stages; the spent catalyst was dismantled by cutting the metal casing and then crushed and the resulting particles were analysed (morphology, particle size distribution and chemical composition). The leaching was undertaken using aqua regia followed by further separation using the functional materials. The main aim of this work was to evaluate the efficiency of the proposed extractants hosted on Merrifield microspheres in extracting platinum group metals from the leachates of spent catalytic converters.

**Figure 1.** Chemical structures of extractants: Triethylenetetramine (TETA), Ethane-1,2-dithiol (SS) and *Bis*((1*H*-benzimidazol-2-yl)methyl)sulfide (NSN).

## 2. Materials and Methods

### 2.1. Reagents and Material

The reagents and materials used in this study, including 2,2'-thiodipropionic acid (97%), *o*-phenylenediamine (99.5%), triethylenetetramine (TETA) (97%), ethane-1,2-dithiol (SS) (90%), Merrifield chloromethylated polystyrene-divinyl benzene resin (capacity [Cl]: 1.2 mmol.g$^{-1}$ resin, 40–60 mesh), *N*,*N*-dimethylformamide (99%), methanol (99%), diethyl ether (99%), hydrochloric acid (37%), nitric acid (70%), ammonia (25%), ethanol (98%), and activated charcoal, were purchased from Sigma-Aldrich, Johannesburg, South Africa. All solvents were purchased from Merck and used as received. The spent catalytic converters were purchased from Auto King Used Spares scrapyard in Markman, Port Elizabeth, South Africa.

### 2.2. Instrumentation

Semi-quantitative X-ray fluorescence (XRF) analysis was carried out using a Bruker S1 Titan XRF analyser using the "Precious metals" mode, was used to identify platinum group metals, base metals and other metals. The X-ray diffraction (XRD) analysis was carried using a Bruker AXS (Karlsruhe, Germany) with a diffractometer, D8 advance with a LynxEye detector (position sensitive detector). The XRD characterization was carried out on materials both before and after treatment. Samples were prepared for Scanning Electron Microscopy (SEM) by coating them in gold using a Balzers' sputtering device. The samples were imaged using a TESCAN Vega TS 5136LM typically operated at 20 kV at a working distance of 20 mm. Elemental analysis of samples using the Bruker energy dispersive spectroscopy (EDS) was determined by using the same procedure as described for SEM analysis, except that no sample surface coating was needed. Before images were taken; the nanofibers were coated with gold to prevent surface charging and to protect the surface material from thermal damage by the electron beam.

A Perkin-Elmer 400 FTIR was used to confirm the presence of the expected functional groups during the synthesis steps. The structure of the ligand was determined by $^1$H NMR spectroscopy on a Bruker AMX 400 MHz NMR spectrometer and reported relative to tetramethylsilane (TMS) δ 0.00. A custom-made glass column with the following dimensions were used for the column (dynamic) studies; 10 cm length, an internal diameter of 3.5 mm and a tip diameter of 1 mm. The metal ions analyses (Rh, Pd, Pt, Ru, Ni, Cr, Mn, Zn, Fe, Al) were carried out with a Perkin Elmer (Avio 200) Inductively Coupled Plasma (ICP) spectrometer equipped with an Optical Emission Spectrometer (OES) as the detector at 343.489 nm for Rh, 203.646 nm for Pt, 363.470 nm for Pd, 240.272 for Ru, 206.200 for Zn, 231.604 nm for Ni, 238.204 nm for Fe, 257.610 nm for Mn, 396.153 nm for AI and 267.716 nm for Cr.

### 2.3. Dismantling and Preparation of Leach Solutions

The honeycomb spent catalytic converter samples (Figure 2A) were dismantled (Figure 2B) and ground into 85 μm particles (mesh) that were used in leaching studies (Figure 2C,D). The petrol (P) and diesel (D) spent catalytic converters were used in this study. 40 mL of aqua regia (3 HCl:1 HNO₃) was added to 1 g of sample (meshed spent catalytic converter powder) and the resulting mixture was stirred while heated at 90 °C for 1 h [42]. The samples were cooled and then filtered to remove the undissolved solids. An orange solution was obtained for the petrol (P) sample (Figure 2E) while a yellow solution was obtained for the diesel (D) sample (Figure 2E). The resulting solutions were diluted and analysed using ICP-OES to verify the content of precious metals and base metals concentrations in the spent catalysts.

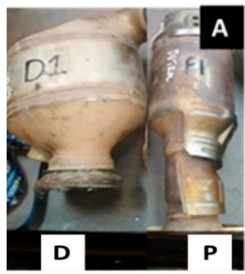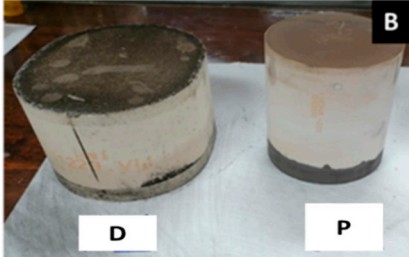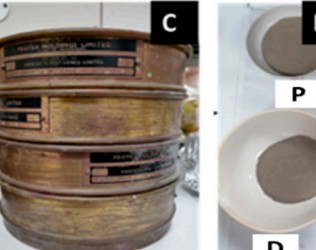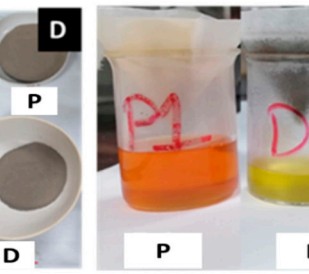

**Figure 2.** Dismantling and leaching process of diesel (D) and petrol (P) spent catalytic converters where (**A**) spent catalytic converters for diesel (D) and petrol (P), (**B**) dismantled catalytic converters, (**C**) Sieving tool used to mesh crushed catalytic converter, (**D**) meshed to up 85 μm particles and (**E**) leached D and P sample using 3:1 HCl: NHO₃.

### 2.4. Leached Metal Solution Analysis

The platinum group metals and base metals standards were prepared in HCl and HNO₃, respectively, for the construction of calibration curves using distilled, deionized water for the dilutions. The metal ion analyses were prepared by measuring 20 μL of the leachate solution and diluting up to 15 mL. The elements, Rh, Pt, Pd, Ru, Zn, Ni, Fe, Mn and Cr, were analysed using ICP-OES.

### 2.5. Synthesis of Bis((1H-benzimidazol-2-yl)methyl)sulfide (NSN)

The synthesis of *bis*((1*H*-benzimidazol-2-yl)methyl)sulfide (NSN) was carried out using a mixture of 2,2'-thiodipropionic (5.0 g, 0.0333 mol) and *o*-phenylenediamine (10.0 g, 0.0930 mol) in 4 M aqueous HCl (250 mL) as shown in Scheme 1 [38–40]. This reaction mixture was refluxed for 24 h. The reaction mixture was then cooled in ice, a precipitate formed and filtered. The precipitate was identified as a dihydrochloride (L.2HCl). The free base ligand was obtained by treatment of the hydrochloride ligand with excess aqueous ammonia. A precipitate formed and was filtered and dissolved in ethanol and decolourized with charcoal. The solution was filtered and concentrated using rotavapor to obtain a light brown solid as a final product and the yield was 75%. Anal. Calcd for C₁₆H₁₄N₄S (%): C, 65.28; H, 4.79; N, 19.03; S, 10.89. Found: C, 64.96; H, 4.58; N, 18.89; S, 10.67. ¹H NMR (400 MHz, DMSO) δ (ppm): 7.53 (4H, s, CH), 7.18 (4H, s, CH), 4.04 (4H, m, CH₂). IR (νmax/cm⁻¹): 3377 ν(N–H), 1534 ν(C=N), 1128 ν(C-S-C).

**Scheme 1.** The synthesis of *bis*((1*H*-benzimidazol-2-yl)methyl)sulfide (NSN).

### 2.6. Preparation of Functionalised Microspheres

The Merrifield resins (M-TETA, M-SS and M-NSN) were prepared by using 3 g of chloromethylated polystyrene beads suspended in 30 mL of DMF and 18 g of each ligand (triethylenetetramine (TETA), 1,2-ethanedithiol (SS) and *bis*((1*H*-benzimidazol-2-yl)methyl)sulfide (NSN)) (Scheme 2). The reaction mixture was stirred for 15 h at 70 °C. The resin was then washed thoroughly with methanol and diethyl ether, and then Soxhlet extraction was carried out using methanol. The resulting functionalised Merrifield beads were analysed using FT-IR (cm$^{-1}$): 1018 m(C–*N*), 1671 d(*N*–H), 3200–3300 m (*N*–H). Anal. found (C, H, *N*%): M-TETA (67.16, 10.47, 22.38), (C, H, S%) M-SS (60.55, 7.11, 32.33) and (C, H, *N*, S%) M-NSN (72.33, 5.56, 14.06, 8.05).

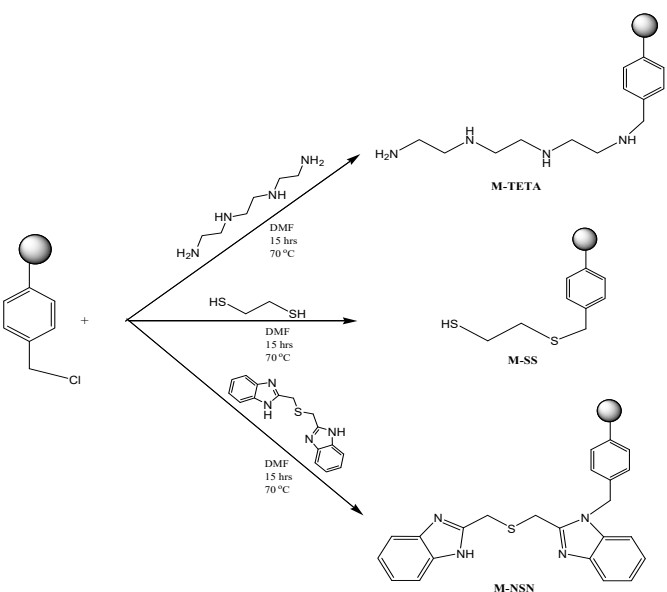

**Scheme 2.** Synthesis scheme for the functionalized Merrifield resins: M-TETA, M-SS and M-NSN.

### 2.7. Column Studies

The capacity of the functionalized Merrifield resins for PGMs (Pt, Pd, Rh and Ru) and base metals (Ni, Mn, Cr, Zn and Fe) uptake from the leachate solutions were determined as a function of fraction number collected in a column. The 0.3 g of resin was conditioned with 5 mL of water followed by 5 mL of 1 M HCl solution. 1.5 mL of a 1 M HCl leachate solution was loaded into the column after the conditioning step. The column was left for 24 h after which it was washed with 5 mL of 1 M HCl to remove metals that did not adhere to the surface of the resin. The extracted metal ions were eluted by collecting 0.5 mL fractions from the loaded resin with 3% thiourea in 1 M HCl through the column. The eluent (0.5 mL fractions) were diluted appropriately and analysed with ICP-OES. The resin loading capacity was calculated as millimoles of metal per gram of the resin (mmol/g).

## 3. Results and Discussion

### 3.1. Catalytic Converters Characterization

The chemical composition of spent automotive catalytic converters for diesel and petrol samples were determined by XRF. Two X-ray fluorescence measurements for diesel and petrol samples have been conducted. It was observed that the catalytic component for the diesel and petrol samples contains a combination of rhodium and platinum; and platinum and palladium, respectively as shown in Table 1. The diesel sample confirmed a higher loading of platinum with 21.11 wt% and petrol higher loading of palladium with 2.90 wt%. PGMs that were not detected by XRF were palladium for the diesel sample and rhodium for the petrol sample. The XRD diffractogram of the spent catalyst before leaching is presented in Figure 3. The XRD result of the petrol sample showed 100% of

the characteristic reflections of the cordierite with the orthorhombic phase $Mg_2(Al_4Si_5O_{18})$ (Figure 3) [43]. The diesel sample showed the cordierite phase as a major component (red) but with minor phases of aluminium phosphate hydrate (turquoise), aluminium oxide (blue), gamma-alumina (light blue), aluminium silicate (green). PGMs phases would be too small to be determined singly by XRD and the cordierite is the dominant phase that would overshadow them.

**Table 1.** The main constituents of the catalysts were obtained with XRF analysis.

| No. | Element (wt%) | Diesel | Petrol | No. | Element (%) | Diesel | Petrol |
|---|---|---|---|---|---|---|---|
| 1 | Si | 31.60 | 1.07 | 13 | Zn | 1.21 | 0.23 |
| 2 | S | 22.93 | ND | 14 | Y | ND | 0.90 |
| 3 | Pt | 21.11 | 0.89 | 15 | Zr | 1.35 | 70.20 |
| 4 | Al | 12.59 | 4.76 | 16 | Mo | 0.07 | 0.34 |
| 5 | Fe | 3.66 | 0.96 | 17 | Pb | ND | 0.04 |
| 6 | V | ND | 11.77 | 18 | Pd | ND | 2.90 |
| 7 | Ti | 1.22 | ND | 19 | In | 0.70 | 0.35 |
| 8 | Cr | 0.87 | 4.69 | 20 | Rh | 0.73 | ND |
| 9 | Mn | 0.15 | 0.49 | 21 | Hf | ND | 1.04 |
| 10 | Cu | 0.85 | ND | 22 | Ta | ND | 1.10 |
| 11 | Co | 0.04 | 0.03 | 23 | W | 0.73 | 0.08 |
| 12 | Ni | 0.13 | 0.10 | | | | |

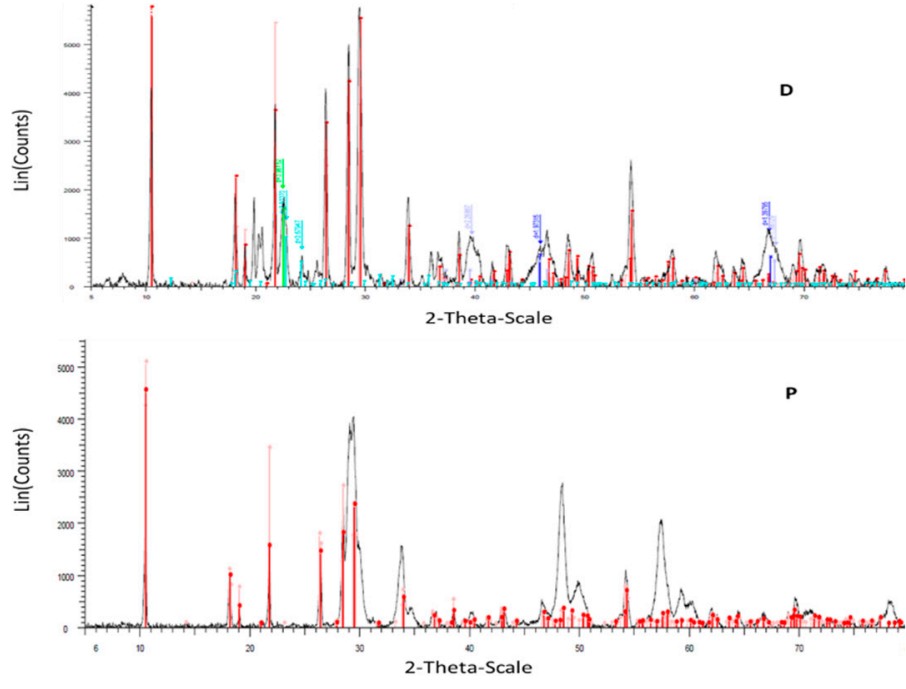

**Figure 3.** XRD pattern powder diffraction pattern for the spent catalyst converters confirming the presence of cordierite in the investigated samples of Diesel (D) and Petrol (P) catalytic converters.

The spent catalytic convertors were dissected for morphology examination for both the diesel and petrol samples. The samples were carefully examined under the Scanning Electron Microscope (SEM) combined with energy-dispersive X-ray spectroscopy analysis (EDS) to provide an additional understanding of the surface material composition. The micrographs obtained corresponding to D and P samples confirmed the composition of

the honeycomb cordierite structure of the catalytic converter as depicted by Figure 4. SEM observation confirmed the presence of PGMs with honeycomb cordierite skeleton-type $(2MgO\text{-}2Al_2O_3\cdot5SiO_2)$. The morphology indicates a single body possessing maltitude parallel channels, with the catalytically active material deposited along the walls of the channels [43]. The energy dispersive x-ray (EDX) analysis on the spent catalytic converters is shown in Figure 5, and the images confirm the presence of Pt, Pd and Rh metals in both diesel and petrol catalysts. The EDX analysis confirmed the presence of other elementals, carbon (C), oxygen (O), magnesium (Mg), aluminium (Al), silicon (Si), phosphorous (P), calcium (Ca), titanium (Ti), molybdenum (Mo), and manganese (Mn) and are all part of essential minerals chiefly found in catalytic converters samples.

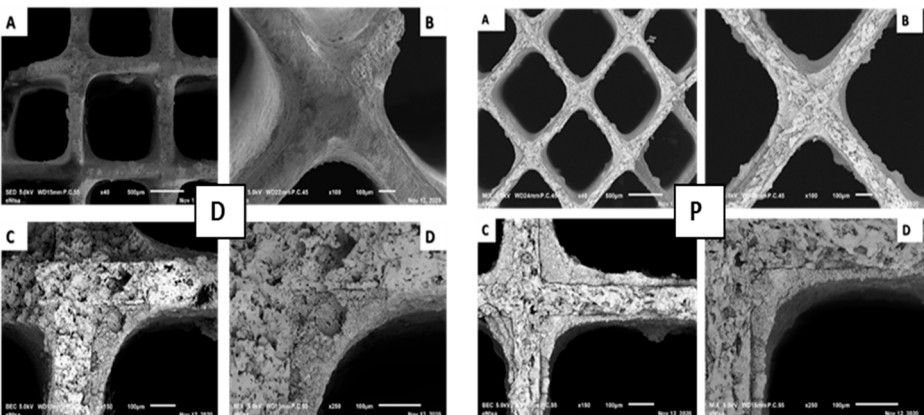

**Figure 4.** Scanning electron micrographs (SEM) of diesel (D) and petrol (P) spent catalytic converters, at different magnifications where ((**A**): 500 μm (×40), (**B**): 100 μm (×100), (**C**): 100 μm (×150) and (**D**): 100 μm (×250)).

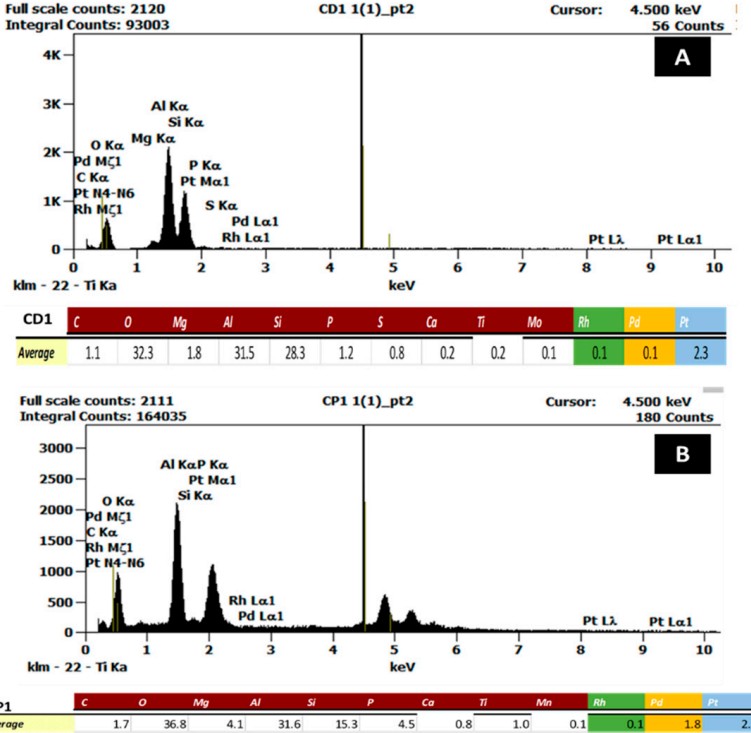

| CD1 | C | O | Mg | Al | Si | P | S | Ca | Ti | Mo | Rh | Pd | Pt |
|---|---|---|---|---|---|---|---|---|---|---|---|---|---|
| Average | 1.1 | 32.3 | 1.8 | 31.5 | 28.3 | 1.2 | 0.8 | 0.2 | 0.2 | 0.1 | 0.1 | 0.1 | 2.3 |

| CP1 | C | O | Mg | Al | Si | P | Ca | Ti | Mn | Rh | Pd | Pt |
|---|---|---|---|---|---|---|---|---|---|---|---|---|
| Average | 1.7 | 36.8 | 4.1 | 31.6 | 15.3 | 4.5 | 0.8 | 1.0 | 0.1 | 0.1 | 1.8 | 2.3 |

**Figure 5.** The energy dispersive x-ray (EDX) analysis images of (**A**) diesel (D) and (**B**) petrol (P) spent catalytical converters.

### 3.2. Leach Solution Analysis

The leaching of PGMs and base metal from diesel and petrol spent catalytic converters followed the long-established leaching system of using a high acidity aqua regia. The ICP-OES was used to confirm the presence of PGMs (Pd, Pt, Rh, Ru) and base metals (Cr, Fe, Mn, Ni, Zn). In this leaching method, the results obtained showed that the percentage of PGMs leaching greatly depended on the leaching solution used at ambient temperature with a 136.97 ppm concentration of Pt for the diesel sample observed compared with 52.37 ppm Pt for the petrol sample. A concentration of 89.40 ppm for Pd was higher for the petrol sample while Rh and Ru were lower in concertation for both petrol and diesel samples (Figure 6). Leaching results based on concentrations as well as %metal based on sample mass used are observed to have the following order: Pt (136.97 ppm, 0.55%) > Pd (27.30 ppm, 0.11%) > Ru (5.71 ppm, 0.023%) > Rh (4.08 ppm, 0.016%) for the diesel sample and Pd (89.40 ppm, 0.36%) > Pt (52.37 ppm, 0.21%) > Rh (9.98 ppm, 0.040%) > Ru (6.53 ppm, 0.026%) for the petrol sample. Cumulatively, this translates to 0.17% PGMs in a diesel sample and 0.16% PGMs in a petrol sample, assuming 100% leaching.

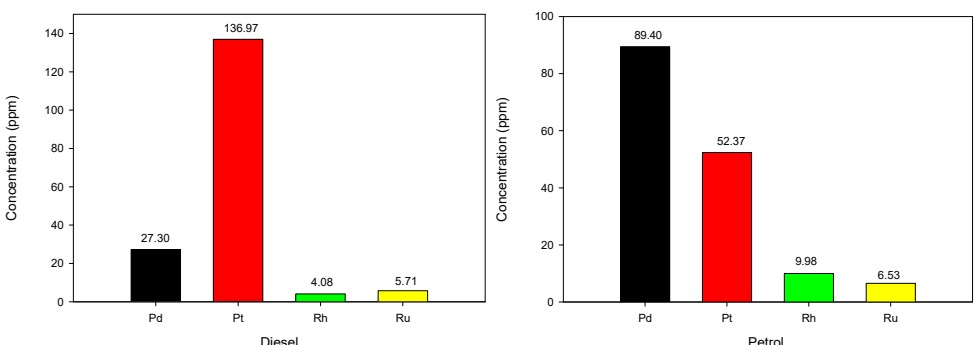

**Figure 6.** Leach solution analysis of platinum group metals from the diesel and petrol spent catalytic converters.

The advantage of using aqua regia produces chlorine, which is an aggressive oxidant [44]. As a result, PGMs dissolved in the form of chlorido complexes $PtCl_6^{2-}$, $PdCl_4^{2-}$ and $RhCl_6^{3-}$ leaving the cordierite substrate as a leach residue [45]

$$3Pt(s) + 18HCl(aq) + 4HNO_3\,(aq) \leftrightarrow 3[PtCl_6]^{2-}(aq) + 6H^+(aq) + 4NO(g) + 8H_2O \qquad (1)$$

$$3Pd(s) + 12HCl(aq) + 2HNO_3(aq) \leftrightarrow 3[PdCl_6]^{2-}(aq) + 6H^+(aq) + 2NO(g) + 4H_2O \qquad (2)$$

$$2Rh(s) + 12HCl(aq) + 2HNO_3(aq) \leftrightarrow 2[RhCl_6]^{3-}(aq) + 6H^+(aq) + 2NO(g) + 4H_2O \qquad (3)$$

Analysis of base metals from leachates was performed to acquire an insight into the dissolution phenomena related to Cr, Fe, Mn, Ni and Zn. From the results obtained, the concentration for Zn (95.37 ppm) was higher in diesel and petrol samples, and Fe (70.43 ppm) was observed to be higher for the diesel sample compared with the petrol sample. The leaching results were achieved in a single step in aqua regia. The efficiency demonstrated that PGMs can be obtained from the leaching of catalytic converters with aqua regia but that base metals are also simultaneously leached. However, some of the cordierite support becomes dissolved in the aqua regia as evidenced by the 26.3–28.0 ppm Al detected in the leach solution which is about 10.5–11.2% of Al in the petrol sample.

### 3.3. Column Studies

The performance of the functionalized Merrified resins M-TETA, M-SS and M-NSN were used for the recovery of PGMs and base metals under dynamic flow adsorption conditions. The multi-element leachate solution containing PGMs (Pt, Pd, Rh, Ru) and base

metals (Cr, Fe, Mn, Ni and Zn) was loaded on the column to study the adsorption/elution profiles of the sorbent materials. The column was washed with 1 M HCl to remove unabsorbed ions (fraction 1–10), followed by elution/stripping with 3% *w/v* thiourea in 1 M HCl solution (fraction 11–50) at ambient temperature. The multi-element column elution for M-TETA, M-SS and M-NSN resins showed an uptake of the Rh, Pt, Pd, Ru and Zn, Ni, Fe, Mn and Cr. The concentration of PGMs and base metals were each fraction collected and determined. The performance of each ligand (TETA, SS and NSN) was evaluated. The loading capacity of the materials for PGMs and base metals was calculated from the total amount of the collected fractions after stripping and calculated as total moles of each metal (mmol)/mass of the polymer material (g) from the concentrations (mg/L) obtained by ICP analysis.

### 3.3.1. Recovery of Platinum Group Metals

Figures 7 and 8 represent the column elution profiles and loading capacities, respectively, for the functionalized resins. The highest loading capacity seems to be for Pd with M-TETA highest followed by M-NSN and then M-SS. It seems that nitrogen content is important for the functioning of the resins [36]. Under acidic conditions, the amine groups function as ammonium sites in ion exchangers to induce the adsorption of the chlorido complexes of PGMs through electrostatic interactions [46]. It is possible that the inner sphere complexation is not dominant in these highly acidic solutions but is expected to be active in M-SS [47]. The loading capacities for Pd decrease in the order M-TETA (192 mmol/g) > M-NSN (177 mmol/g) > M-SS (64 mmol/g) and this order is reproduced for the other metal ions albeit in lower quantities (Table 2). The M-SS and M-NSN were, respectively, taken through a second cycle and the recoveries of specific metals are somewhat consistent (Table 2). For example, Cycle 1 for M-NSN resin gave a Pd recovery of 177 mmol/g and Cycle 2 gave 131 mmol/g.

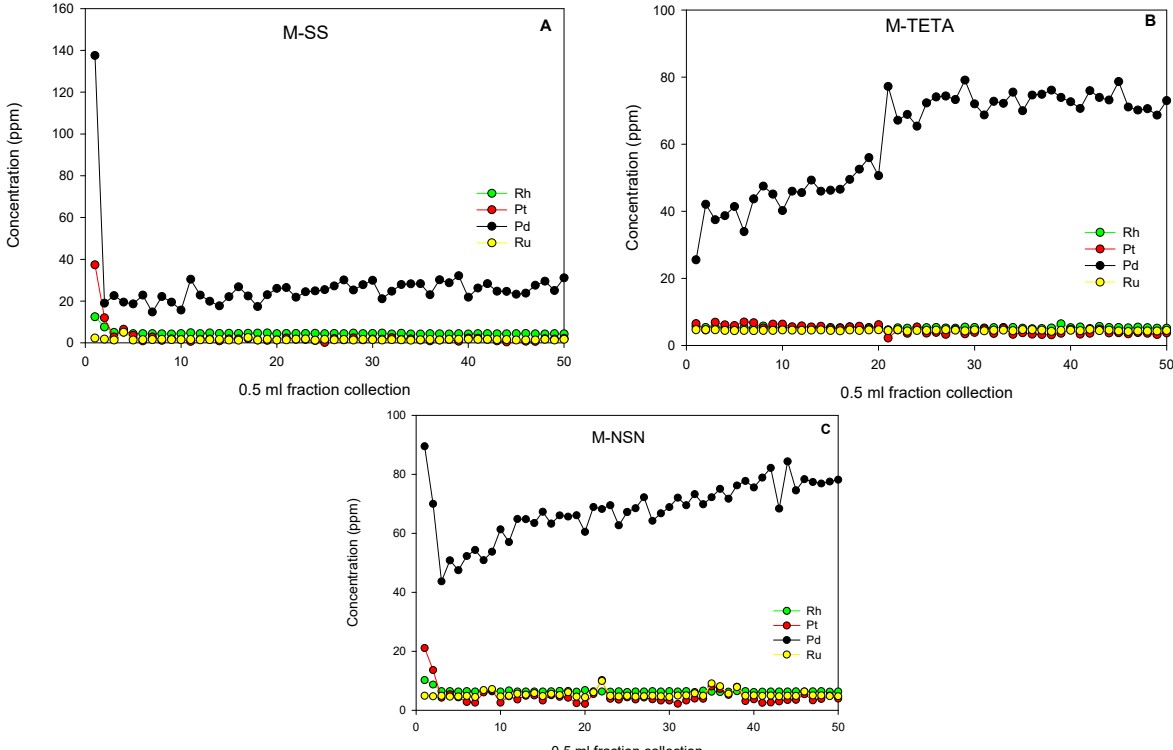

**Figure 7.** PGMs adsorption/elution profiles using 5 mL of 1M of the leachate solution, 0.3 g resins: (**A**) M-TETA, (**B**) M-SS and (**C**) M-NSN were, respectively, washed with 1 M HCl and eluted with 3% *w/v* thiourea at ambient temperature.

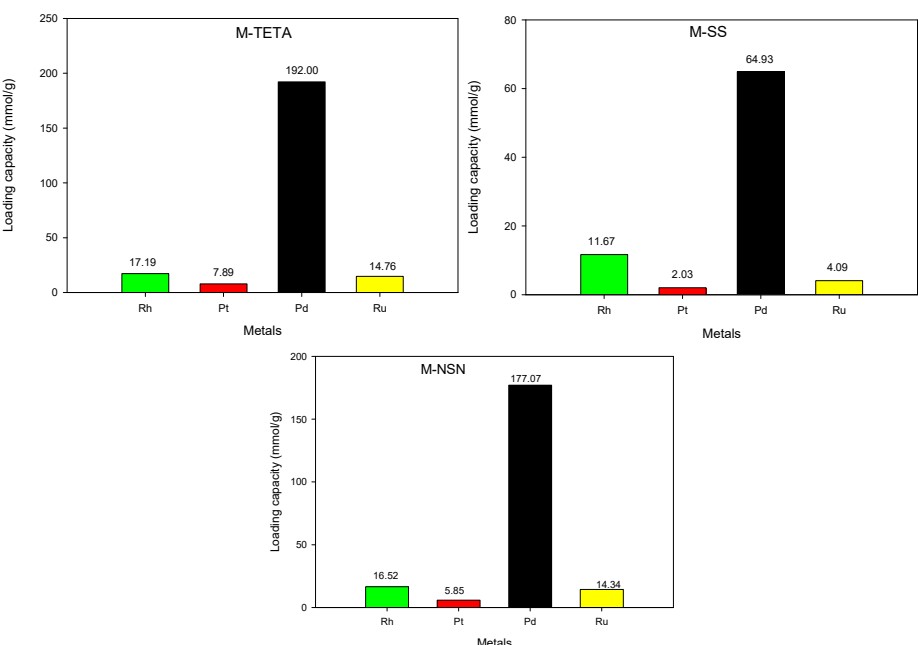

**Figure 8.** Loading capacities of the functionalized Merrifield resins for Pt, Pd, Rh and Ru at a flow rate of 0.5 mL/h for Cycle 1.

**Table 2.** Loading capacities of platinum group metals extracted using functionalized Merrifield resins M-TETA, M-SS, and M-NSN.

| Merrifield Resin | Rh (mmol/g) | | Pt (mmol/g) | | Pd (mmol/g) | | Ru (mmol/g) | |
|---|---|---|---|---|---|---|---|---|
| | Cycle 1 | Cycle 2 | Cycle 1 | Cycle 2 | Cycle 1 | Cycle 2 | Cycle 1 | Cycle 2 |
| M-TETA | 17.19 | - * | 7.88 | - * | 192.00 | - * | 14.76 | - * |
| M-SS | 11.67 | 9.03 | 2.03 | 3.40 | 64.93 | 60.54 | 4.09 | 5.24 |
| M-NSN | 16.52 | 9.12 | 5.85 | 3.79 | 177.07 | 131.01 | 14.34 | 5.078 |

\* M-TETA 2nd cycle was not determined.

### 3.3.2. Recovery of Base Metals

The possibility of simultaneous recovery of base metals was investigated to evaluate the practical applications of the resin for the recovery of PGMs. From the same solution loaded for the recovery of PGMs, the base metals were analysed for each sorbent and the extraction of base metal was observed (Figure 9). The recovery of Ni and Mn was observed in all three resins as shown in Table 3. The M-SS and M-NSN sorbents could recover all base metal under study (Zn, Ni, Fe, Mn and Cr) with higher loading capacity for Zn (31.77 mmol/g) for M-SS and a higher loading capacity of Fe (489.55 mmol/g) for M-NSN. M-TETA was unable to recover Zn and Fe but could recover the other three base metals (Ni (19.49 mmol/g), Mn (29.10 mmol/g), Cr (14.49 mmol/g). The highest recovery for base metals was achieved by M-NSN indicating that the recovery of base metals is favoured by a greater preference for π acceptor ligands [38]. As discussed in the PGMs recovery section, the high recovery of PGMs attained with M-TETA is accompanied by a low recovery of base metals. These results demonstrated the importance of the chemistry of the resin in its functioning to recover specific metals. In this case, selective separations were not fully achieved but the simultaneous recovery of PGMs and exclusion of some based metals by M-TETA is promising.

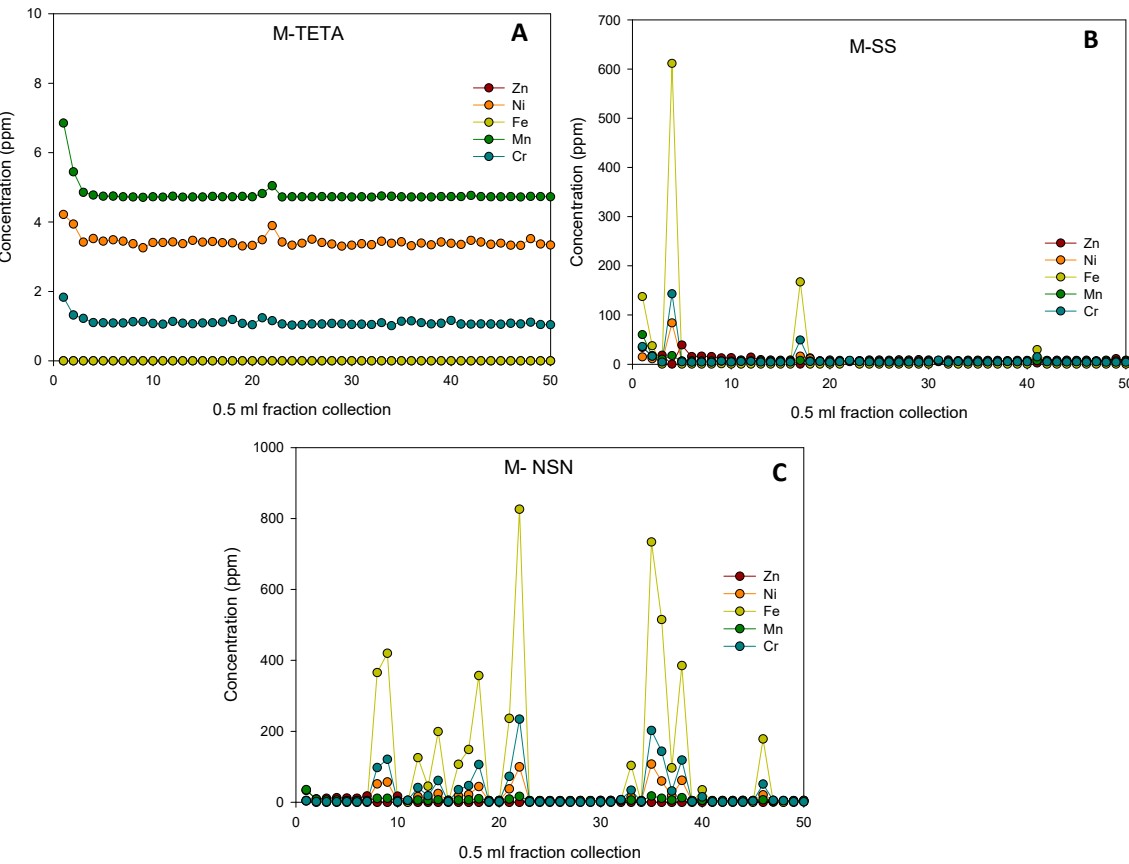

**Figure 9.** Base metals adsorption/elution profiles using 5 mL of 1 M leachate solution on 0.3 g functionalized Merrifield resins: (**A**) M-TETA, (**B**) M-SS and (**C**) M-NSN were, respectively, washed with 1 M HCl and eluted with 3% *w/v* thiourea at ambient temperature.

**Table 3.** Loading capacities of base metals extracted using microspheres beads functionalized M-TETA, M-SS, and M-NSN.

| Functionalized Microspheres | Zn (mmol/g) | Ni (mmol/g) | Fe (mmol/g) | Mn (mmol/g) | Cr (mmol/g) |
|---|---|---|---|---|---|
| M-TETA | - | 19.49 | - | 29.10 | 14.49 |
| M-SS | 37.77 | 26.82 | 25.25 | 30.45 | 26.57 |
| M-NSN | 9.02 | 74.91 | 489.55 | 29.64 | 140.34 |

## 4. Conclusions

This study demonstrates an interesting case of functionalized Merrifield resins for the recovery of platinum group metals and base metals from leachates of spent catalytic converters. The hydrometallurgical process for PGMs recovery from spent catalytic converter was followed which included: dismantling, leaching with mineral acids, extraction of platinum group metals and lastly recovery of base metals. The leaching of PGMs using aqua regia was successful and this confirmed the cumulative 0.1–0.2% quantity of PGMs in spent catalytic converters. The recovery of PGMs from leach solutions using M-TETA, M-SS and M-NSN was also successful albeit with simultaneous recovery of base metals. M-TETA showed the highest uptake of palladium followed by M-NSN and then M-SS. Sulfur ligands are known the convert to sulfoxides in highly oxidizing media, hence it is possible that the S-donor ability of M-SS was partially inactivated. The order for the uptake of other PGMs was similar with respect to the resins albeit much lower quantities were extracted. The highest quantities obtained were for palladium with all three resins and

one can possibly consider these resins as having a higher affinity for palladium. Nevertheless, the simultaneous recovery of PGMs and base metals is the only limitation for their palladium selectivity and future studies require further ligand design strategies around these functional groups to eliminate the co-extraction of smaller quantities of PGMs and base metals.

**Author Contributions:** Z.R.T., conceptualization, supervision, data analysis and editing; P.M.-B., data collection, data analysis, writing, and editing; M.Z.N., data collection and data analysis; H.M., data analysis, writing, and editing. All authors have read and agreed to the published version of the manuscript.

**Funding:** This research was funded by the National Research Foundation (NRF) of South Africa (Grant number 129274). NRF is also acknowledged for funding P. Moleko-Boyce (Postdoctoral Scholarship; reference no: SFP180425324247; unique grant no: 116726). Nelson Mandela University Research Capacity Development is acknowledged for financial assistance towards a postdoctoral fellow funding for Hlamulo Makelane.

**Data Availability Statement:** Data is available from authors.

**Acknowledgments:** The authors acknowledge Nelson Mandela University for providing research facilities.

**Conflicts of Interest:** There is no conflict of interest for this work.

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
