# Peer review of "Recovery of Platinum Group Metals from Leach Solutions of Spent Catalytic Converters Using Custom-Made Resins"

_minerals, doi:10.3390/min12030361_

Round 1

Reviewer 1 Report

I have reviewed article minerals-1537722, "Recovery of Precious Metals from Spent Catalytic Converters" by P. Moleko-Boyce, H. Makelane and Z.R. Tshentu. The topic is interesting and promising and important from the point of view of the circular economy. Natural sources of PGM are very poor, which prompts us to obtain PGM from secondary sources (e.g. spent automotive catalysts).

Comments and suggestions which authors may find useful in upgrading manuscript are the following:

  1. Title of the manuscript: Looking at the title of the manuscript, I thought it would be a review article. I suggest to do small change of the title, because the title of the publication is a bit too general.
  2. Lines 51-52: From the literature and my own research, the content of PGM in the spent automotive converters is even about 1%. I suggest the authors to check also in another sources (literature).
  3. Chapter 2.2. Instrumentation - please give the names and producers of the used apparatus (XRF, XRD, EDS).
  4. In my opinion, Figure 2 should be moved to the appendix A.
  5. Lines 203-206, Table 1, Figure 5: The authors confirmed the presence of cordierite. Cordierite is a magnesium aluminum silicate and can contain up to 35% aluminum. So why was aluminum not determined in the samples after leaching?
  6. Figures 6, 8 and 9 - Please use the same color to the same element, e.g. in figure 6 the bar for Pd is black but in figure 9 the bar for Pd is green (black barn is for Rh).
  7. Can sorbent materials be used again?
  8. Other comments:
  • All text: In my opinion, “non-precious metals” should be used instead of “base metals”,
  • Lines 50, 56: is “NOx” but it should be “NOx”,
  • Lines 238, 266, 284, 307: is “room temperature” but it should be “ambient temperature”,
  • Equations 1-3: is “H+” but it should be “H+”.

Reviewer 2 Report

The authors presented the important topic of recovering platinum group metals from spent catalytic converters. The article is well written, but the authors must clarify certain issues and introduce editorial changes - please read and refer to the list of comments below.  

  • the article requires an English correction - errors have been noticed, e.g. line 10 - it should be "efforts are", line 16 - it should be "for recovery" etc. 
  • Figure 2: under the title of the figure, explanations of the symbols A, B, C etc. should be added. 
  • What sets the leaching experiments in this article apart from other publications? The manuscript presents the methodology and results of leaching using aqua regia as the leaching agent, which is widely used in the literature and is not a novelty in the recovery of precious metals from waste.
  • Figure 3: main XRD peaks should be described - what compounds/elements they represent.
  • Section 3.2: in the literature, leaching results are usually presented as recovery rates (in %) of individual metals relative to their initial content in the test material, which is easier for the reader to understand, and the process efficiency can be compared.
  • Conclusions: should be more elaborate, presenting a summary of the entire research part and final conclusions (e.g. there is no emphasis on the validity of research and the importance of its results). 
